# SGD Converges to Global Minimum in Deep Learning via Star-convex Path

**Yi Zhou**[*], **Junjie Yang**[†], **Huishuai Zhang**[‡], **Yingbin Liang**[§], **Vahid Tarokh**[*]
[*]Duke University, [†]University of Science and Technology of China
[‡]Microsoft Research, Asia, [§]The Ohio State University

## Abstract

Stochastic gradient descent (SGD) has been found to be surprisingly effective in training a variety of deep neural networks. However, there is still a lack of understanding on how and why SGD can train these complex networks towards a global minimum. In this study, we establish the convergence of SGD to a global minimum for nonconvex optimization problems that are commonly encountered in neural network training. Our argument exploits the following two important properties: 1) the training loss can achieve zero value (approximately), which has been widely observed in deep learning; 2) SGD follows a star-convex path, which is verified by various experiments in this paper. In such a context, our analysis shows that SGD, although has long been considered as a randomized algorithm, converges in an intrinsically deterministic manner to a global minimum.

## 1 Introduction

Training neural networks has been proven to be NP-hard decades ago Blum & Rivest (1988). At that time, the limited computation power makes neural network training a "mission impossible". However, as the development of computing device achieves several revolutionary milestones (e.g. GPUs), deep neural networks are found to be practically trainable and can generalize well on real datasets Krizhevsky et al. (2017). At the same time, deep learning technique starts to beat the performance of other conventional approaches in a variety of challenging tasks, e.g., computer vision, classification, natural language processing, etc.

Modern neural network training is typically performed by applying first-order algorithms such as stochastic gradient descent (SGD) (a.k.a backpropagation) (Linnainmaa, 1976) or its variants, e.g., Adam (Kingma & Ba, 2014), Adagrad (Duchi et al., 2011), etc. Traditional analysis of SGD in nonconvex optimization guarantees the convergence to a stationary point Bottou et al. (2016); Ghadimi et al. (2016). Recently, it has been shown that SGD has the capability to escape strict saddle points Ge et al. (2015); Jin et al. (2017); Reddi et al. (2018a); Daneshmand et al. (2018), and can escape even sharp local minima Kleinberg et al. (2018). While these works provide different insights towards understanding the performance of SGD, they cannot explain the success of SGD that has been widely observed in deep learning applications. Specifically, it is known that SGD is able to train a variety of deep neural networks to achieve zero training loss (either exactly or approximately) for non-negative loss functions. This implies that SGD can find a *global* minimum of deep neural networks at ease. The major challenges towards understanding this phenomenon are in two-fold: 1) deep neural networks have complex landscape that cannot be fully understood analytically (Zhou & Liang, 2018); 2) the randomness nature of SGD makes it hard to characterize its convergence on such a complex landscape. These factors prohibit a good understanding of the practical success of SGD in deep learning from a traditional optimization aspect.

In this study, we analyze the convergence of SGD in the above phenomenon by exploiting the following two critical properties. First, the fact that SGD can train neural networks to zero loss value implies that the non-negative loss functions on all data samples share a common global minimum. Second, our experiments establish strong empirical evidences that SGD (when training the loss to zero value) follows a star-convex path. Based on these properties, we formally establish the convergence of SGD to a global minimizer. Our work conveys a useful insight that although the landscape of

neural networks can be complicated, the actual optimization path that SGD takes turns out to be remarkably simple and sufficient to guarantee the converge to a global minimum.

## 1.1 OUR CONTRIBUTIONS

We focus on the empirical observation that SGD can train various neural networks to achieve zero loss, and validate that SGD follows an epochwise star-convex path in empirical optimization processes. Based on such a property, we prove that the Euclidean distance between the variable sequence generated by SGD and a global minimizer decreases at an epoch level. We also show that the subsequences of iterations that correspond to the same data sample is a minimizing sequence of the loss corresponding to that sample.

By further empirical exploration, we validate that SGD follows an iterationwise star-convex path during the major part of the training process. Based on such a property, we prove that the entire variable sequence generated by SGD converges to a global minimizer of the objective function. Then, we show that the convergence of SGD induces a self-regularization on its variance, i.e., the variance of stochastic gradients vanishes as SGD converges in such a context.

From a technical perspective, we characterize the intrinsic deterministic convergence property of SGD when the optimization path is well regularized, rather than the performance on average or in probability established in the existing studies. Our results provide a novel and promising aspect to understand SGD-based optimization in deep learning. Furthermore, our analysis of SGD explores the limiting convergence property of the subsequences that correspond to individual data samples, which is in sharp contrast to the traditional treatment of SGD that depends on its random nature and bounds on variance. Hence, our proof technique can be of independent interest to the community.

## 1.2 RELATED WORK

As there are extensive literature on SGD, we only mention the highly relevant studies here. Theoretical foundations of SGD have been developed in the optimization community (Robbins & Monro, 1951; Nemirovski et al., 2009; Lan, 2012; Ghadimi et al., 2016; Ghadimi & Lan, 2016), and have attracted much attention from the machine learning community in the past decade (Schmidt et al., 2017; Defazio et al., 2014; Johnson & Zhang, 2013; Li et al., 2017; Wang et al., 2018a). In general nonconvex optimization, it is known that SGD converges to a stationary point under a bounded variance assumption and a diminishing learning rate Bottou et al. (2016); Ghadimi et al. (2016). Other variants of SGD that are designed for deep learning have been proposed, e.g., Adam (Kingma & Ba, 2014), AMSgrad Reddi et al. (2018b), Adagrad (Duchi et al., 2011), etc, and their convergence properties have been studied in the context of online convex regret minimization.

Needell et al. (2014); Moulines & Bach (2011) show that SGD converges at a linear rate when the objective function is strongly convex and has a unique common global minimum. Recently, several studies show that SGD has the capability to escape strict saddle points Ge et al. (2015); Jin et al. (2017); Reddi et al. (2018a); Daneshmand et al. (2018). Other cubic-regularization-based methods have also been shown to be able to escape strict saddle points Nesterov & Polyak (2006); Zhou et al. (2018); Wang et al. (2018b). Kleinberg et al. (2018) considers functions with one-point strong convexity, and shows that the randomness of SGD has an intrinsic smoothing effect that can avoid convergence to sharp minimum. Our paper exploits a very different notion of star-convexity path of SGD, which is a much weaker condition than those in the previous studies.

## 2 PROBLEM SETUP AND PRELIMINARIES

Neural network training can be formulated as the following finite-sum optimization problem.

$$\min_{x \in \mathbb{R}^d} f(x) := \frac{1}{n} \sum_{i=1}^n \ell_i(x), \tag{P}$$

where there are in total $n$ training data samples. The loss function that corresponds to the $i$-th data sample is denoted by $\ell_i : \mathbb{R}^d \to \mathbb{R}$ for $i = 1, \ldots, n$, and the vector to be minimized in the problem (e.g., network weights in deep learning) are denoted by $x \in \mathbb{R}^d$. In general, problem (P) is a

nonconvex optimization problem, and we make the following standard assumptions regarding the objective function.

**Assumption 1.** *The loss functions $\ell_i, i = 1, \ldots, n$ in problem (P) satisfy:*

*1. They are continuously differentiable, and their gradients are L-Lipschitz continuous;*

*2. For every $i = 1, \ldots, n$, $\inf_{x \in \mathbb{R}^d} \ell_i(x) > 0$.*

The conditions imposed by Assumption 1 are standard in analysis of nonconvex optimization. In specific, item 1 is a standard smoothness assumption on nonconvex loss functions. Item 2 assumes that the loss functions are bounded below, which is satisfied by many loss functions in deep learning, e.g., MSE loss, crossentropy loss, NLL loss, etc, which are all non-negative.

Next, we introduce the following fact that is widely observed in training overparameterized deep neural networks, which we assume to hold throughout the paper.

**Observation 1** (Global minimum in deep learning). *The objective function $f$ in problem (P) with non-negative loss can achieve zero value at certain $x^*$. Thus, $x^*$ is also a common global minimizer for all individual loss $\{\ell_i\}_{i=1}^n$. More formally, denote $\mathcal{X}_i^*$ as the set of global minimizers of $\ell_i$ for $i = 1, \ldots, n$. Then, the set of common global minimizers, i.e., $\mathcal{X}^* := \cap_{i=1}^n \mathcal{X}_i^*$, is non-empty and bounded.*

Observation 1 is a common observation in deep learning applications, because deep neural networks (especially in the overparameterized regime) typically have enough capacity to fit all training data samples, and therefore the model has a global minimum shared by the loss functions on all data samples. To elaborate more, if the total loss $f$ attains zero value at $x^*$, then $\ell_i$ for all $i$ must achieve zero value at $x^*$ as they are non-negative. Thus, $x^*$ is a common global minimum of all individual loss $\ell_i(x)$ for all $i$. Such a fact plays a critical role in understanding the convergence of SGD in training neural networks.

Next, we introduce our algorithm of interest – stochastic gradient descent (SGD). Specifically, to solve problem (P), SGD starts at an initial vector $x_0 \in \mathbb{R}^d$ and generates a variable sequence $\{x_k\}_k$ according to the following update rule.

$$(\text{SGD}) \quad x_{k+1} = x_k - \eta \nabla \ell_{\xi_k}(x_k), \quad k = 0, 1, \ldots,$$

where $\eta > 0$ denotes the learning rate, and $\xi_k \in \{1, \ldots, n\}$ corresponds to the sampled data index at iteration $k$. In this study, we consider the practical cyclic sampling scheme with reshuffle (also referred to as the random sampling without replacement) to generate the random variable $\xi_k$. To elaborate the notation, we rewrite every iteration number $k$ as $nB + t$, where $B = 0, 1, 2, \ldots$ denotes the index of epoch that iteration $k$ belongs to and $t \in \{0, 1, \ldots, n - 1\}$ denotes the corresponding iteration index in that epoch. We further denote $\pi_B$ as the random permutation of $1, \ldots, n$ in the $B$-th epoch, and denote $\pi_B(j)$ as its $j$-th element. Then, the sampled data index at iteration $k$ can be expressed as

$$(\text{Cyclic sampling with reshuffle}): \quad \xi_k = \pi_B(t + 1), \quad t = 0, \ldots, n - 1.$$

## 3 APPROACHING GLOBAL MINIMUM EPOCHWISELY

### 3.1 AN INTERESTING EMPIRICAL OBSERVATION OF SGD PATH

In this subsection, we provide empirical observations on the algorithm path of SGD in training neural networks. To be specific, we train a standard multi-layer perceptron (MLP) network Krizhevsky (2009), a variant of Alexnet and a variant of Inception network Zhang et al. (2017a) on the CIFAR10 dataset Krizhevsky (2009) using SGD under crossentropy loss. In all experiments, we adopt a constant learning rate (0.01 for MLP and Alexnet, 0.1 for Inception) and a constant mini-batch size 128. We discard all other optimization features such as momentum, weight decay, dropout and batch normalization, etc, in order to observe the essential property of SGD.

In each experiment, we train the network for a sufficient number of epochs to achieve near-zero training loss (i.e., almost global minimum), and record the weight parameters along the iteration path of SGD. We denote the weight parameters produced by SGD in the last iteration as $x^*$, which has

a near zero loss, and evaluate the Euclidean distance between the weight parameters produced by SGD and $x^*$ along the iteration path. We plot the results in Figure 1. It can be seen that the training losses for all three networks fluctuate along the iteration path, implying that the algorithm passes through complex landscapes. However, the Euclidean distance between the weight parameters and the final output $x^*$ is monotonically decreasing epochwise along the SGD path for all three networks. This shows that the variable sequence generated by SGD approaches the global minimum $x^*$ in a remarkably stable way. This motivates us to explore the underlying mechanism that yields such interesting observations.

In the next two subsections, we first propose a property that the algorithm path of SGD satisfies, based on which we formally prove that the variable sequence generated by SGD admits the behavior observed in Figure 1. Then, we provide empirical evidences to validate such a property of SGD path in practical SGD training.

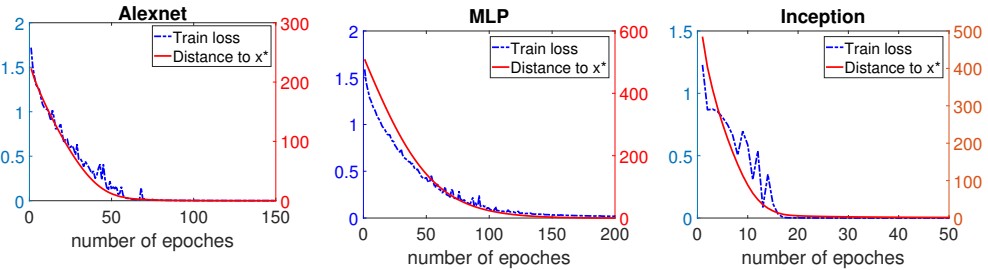

Figure 1: Distance to output of SGD in training neural networks.

## 3.2 Epochwise Star-convex Path

In this subsection, we introduce the notion of the epochwise star-convex path for SGD and establish its theoretical implications on the convergence of SGD. We validate that SGD satisfies such a property in practical neural network training in Section 3.3.

Recall the conventional definition of star-convexity. Let $x^*$ be a global minimizer of a smooth function $h$. Then, $h$ is said to be star-convex at a point $x$ provided that

$$(\text{Star-convexity}): \quad h(x) - h(x^*) + \langle x^* - x, \nabla h(x) \rangle \leq 0.$$

Star-convexity can be intuitively understood as convexity between a reference point $x$ and a global minimizer $x^*$. Such a property ensures that the negative gradient $-\nabla h(x)$ points to the desired direction $x^* - x$ for minimization.

Next, we define the notion of epochwise star-convex path, which requires the star-convexity to be held cumulatively over each epoch.

**Definition 1** (Epochwise star-convex path). *We call a path generated by SGD epochwise star-convex if it satisfies: For all epochs $B = 0, 1, ...$ and for a fixed $x^* \in \mathcal{X}^*$ (see Observation 1 for definition),*

$$\sum_{k=nB}^{n(B+1)-1} \left[ \ell_{\xi_k}(x_k) - \ell_{\xi_k}(x^*) + \langle x^* - x_k, \nabla \ell_{\xi_k}(x_k) \rangle \right] \leq 0.$$

We note that the property introduced by Definition 1 is not about the landscape geometry of a loss function, which can be complex as observed in the training loss curves shown in Figure 1. Rather, it characterizes the interaction between the algorithm and the loss function along the optimization path. Such a property is generally weaker than the global star-convexity, and is observed to be held in practical neural network training (see Section 3.3).

Based on Definition 1, we obtain the following property of SGD.

**Theorem 1** (Epochwise diminishing distance). *Let Assumption 1 hold and apply SGD with learning rate $\eta < \frac{1}{L}$ to solve problem (P). Assume SGD follows an epochwise star-convex path for a certain $x^* \in \mathcal{X}^*$. Then, the variable sequence $\{x_k\}_k$ generated by SGD satisfies, for all epochs $B = 0, 1, ...$,*

$$\|x_{n(B+1)} - x^*\| \leq \|x_{nB} - x^*\|. \tag{1}$$

Theorem 1 proves that the variable sequence generated by SGD approaches a global minimizer at an epoch level, which is consistent with the empirical observations made in Figure 1. Therefore, the property of epochwise star-convex path of SGD is sufficient to explain such desirable empirical observations, although the loss function can be highly nonconvex and has complex landscape.

Under the cyclic sampling scheme with reshuffle, SGD samples every data sample once per epoch. Consider the loss $\ell_v$ on the $v$-th data sample for a fixed $v \in \{1, 2, ..., n\}$. One can check that the iterations in which the loss $\ell_v$ is sampled form a subsequence $\{x_{nB + \pi_B^{-1}(v)}\}_B$, where $\pi_B^{-1}$ is the inverse permutation mapping of $\pi_B$, i.e., $\pi_B^{-1}(u) = v$ if and only if $\pi_B(v) = u$. Next, we characterize the convergence properties of these subsequences corresponding to the loss functions on individual data samples.

**Theorem 2** (Minimizing subsequences). *Under the same settings as those of Theorem 1, the subsequences $\{x_{nB + \pi_B^{-1}(v)}\}_B$ for $v = 1, ..., n$ satisfy*

*1. They are minimizing sequences for the corresponding loss functions, i.e.,*

$$\lim_{B \to \infty} \ell_v(x_{nB + \pi_B^{-1}(v)}) = \inf_{x \in \mathbb{R}^d} \ell_v(x), \quad \forall v \in \{1, ..., n\}.$$

*2. Every limit point of $\{x_{nB + \pi_B^{-1}(v)}\}_B$ is in $\mathcal{X}_v^*$.*

Theorem 2 characterizes the limiting behavior of the subsequences that correspond to the loss functions on individual data samples. Essentially, the results in items 1 and 2 show that each subsequence $\{x_{nB + \pi_B^{-1}(v)}\}_B$ is a minimizing sequence for the corresponding loss $\ell_v$.

### DISCUSSION

We note that Theorems 1 and 2 characterize the epochwise convergence property of SGD in a deterministic way. The underlying technical reason is that the common global minimizer structure in Observation 1 suppresses the randomness induced by sampling and reshuffling of SGD, and ensures a common direction along which SGD can approach the global minimum on all individual data samples. Such a result is very different from traditional understanding of SGD where randomness and variance play a central role Nemirovski et al. (2009); Ghadimi et al. (2016).

### 3.3 VERIFYING EPOCHWISE STAR-CONVEX PATH OF SGD

In this subsection, we conduct experiments to validate the epochwise star-convex path of SGD introduced in Definition 1.

We train the aforementioned three types of neural networks, i.e., MLP, Alexnet and Inception, on CIFAR10 Krizhevsky (2009) and MNIST Lecun et al. (1998) dataset using SGD. The hyperparameter settings are the same as those mentioned in Section 3.1. We train these networks for a sufficient number of epochs to achieve a near-zero training loss (i.e., near-global minimum). We record the variable sequence generated by SGD along the entire algorithm path, and set $x^*$ to be the final output of SGD. Then, we evaluate the value of the summation term in Definition 1 for each epoch. The value of this summation term for each epoch $B$ is denoted as residual $e_B$. By Definition 1, SGD path in the $B$-th epoch is epochwise star-convex provided that $e_B < 0$.

Figure 2 shows the results of our experiments. In all subfigures, the red horizontal curve denotes the zero value baseline, and the other curve denotes the residual $e_B$. It can be seen from Figure 2 that, on the MNIST dataset (second row), the entire path of SGD satisfies epochwise star-convexity for all three networks. On the CIFAR10 dataset (first row), we observe an epochwise star-convex path of SGD after several epochs of the initial phase of training. This can be due to the more complex landscape of the loss function on the CIFAR10 dataset, so that it takes SGD several epochs to enter a basin of attraction of the global minimum.

Our empirical findings strongly support the validity of the epochwise star-convex path of SGD in Definition 1. Therefore, Theorem 1 establishes an empirically-verified theory for characterizing the convergence property of SGD in training neural networks at an epoch level. In particular, it is well justified to successfully explain the stable epochwise convergence behavior observed in Figure 1.

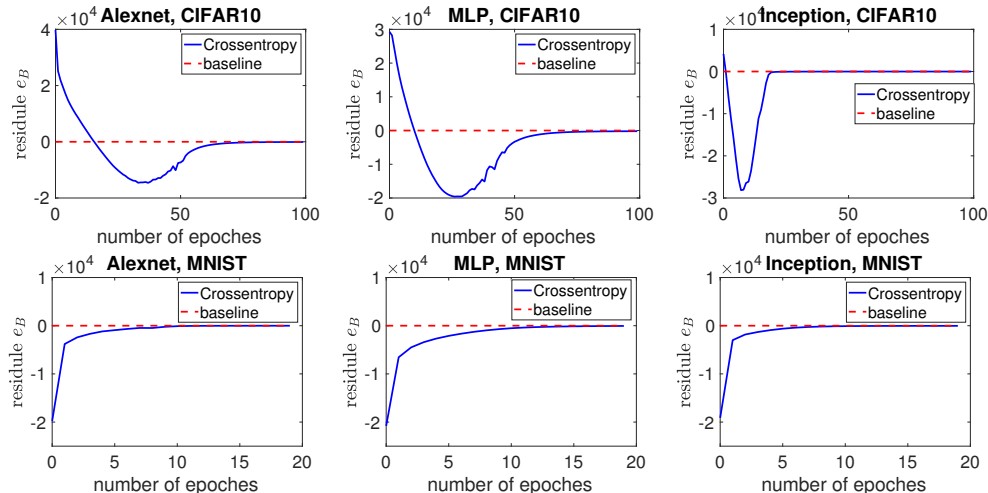

Figure 2: Verification of epochwise star-convex path.

# 4 CONVERGENCE TO A GLOBAL MINIMIZER

The result developed in Theorem 1 shows that the variable sequence generated by SGD monotonically approaches a global minimizer at an epoch level. However, it does not guarantee the convergence of the variable sequence to a global minimizer (which requires the distance between SGD iterates and the global minimizer reduces to zero). We further explore such a convergence issue in the following two subsections. We first define a notion of an iterationwise star-convex path for SGD, based on which we formally establish the convergence of SGD to a global minimizer. Then, we provide empirical evidences to support the satisfaction of the iterationwise star-convex path by SGD.

## 4.1 ITERATIONWISE STAR-CONVEX PATH

We introduce the following definition of an iterationwise star-convex path for SGD.

**Definition 2** (Iterationwise star-convex path). *We call a path generated by SGD iterationwise star-convex if it satisfies: For all $k = 0, 1, ...$ and for every $x^* \in \mathcal{X}^*_{\xi_k}$,*

$$\ell_{\xi_k}(x_k) - \ell_{\xi_k}(x^*) + \langle x^* - x_k, \nabla \ell_{\xi_k}(x_k) \rangle \leq 0. \tag{2}$$

Compared to Definition 1 which defines the star-convex path of SGD at an epoch level, Definition 2 characterizes the star-convexity of SGD along the optimization path at a more refined iteration level. As we show in the result below, such a stronger property helps to regularize the convergence property of SGD at an iteration level, and is sufficient to guarantee convergence.

**Theorem 3** (Convergence to global minimizer). *Let Assumption 1 hold and apply SGD with learning rate $\eta < \frac{1}{L}$ to solve problem (P). Assume SGD follows an iterationwise star-convex path. Then, the sequence $\{x_k\}_k$ generated by SGD converges to a global minimizer.*

Theorem 3 formally establishes the convergence of SGD to a global minimizer along an iterationwise star-convex path. The main idea of the proof is to establish a consensus of the minimizing subsequences that are studied in Theorem 2, i.e., all these subsequences converge to the same limit – a common global minimizer of the loss functions over all the data samples. More specifically, our proof strategy consists of three steps: 1) show that every limit point of each subsequence is a common global minimizer; 2) prove that each subsequence has a unique limit point; 3) show that all these subsequences share the same unique limit point, which is a common global minimizer. We believe that the proof technique here can be of independent interest to the community.

Our analysis in Theorem 3 characterizes the intrinsic deterministic convergence property of SGD, which is an alternative view of the SGD path: It performs gradient descent on an individual loss component at each iteration. The star-convexity along the iteration path pushes the algorithm towards

the common global minimizer. Such progress is shared across all data samples in every iteration and eventually leads to the convergence of SGD.

We also note that the convergence result in Theorem 3 is based on a constant learning rate, which is typically used in practical training. This is very different from and much more desirable than the diminishing learning rate adopted in traditional analysis of SGD Nemirovski et al. (2009), which is a necessity to mitigate the negative effects caused by the variance of SGD. Furthermore, Theorem 3 shows that SGD converges to a common global minimizer where the gradient of loss function on all data samples vanish, and we therefore obtain the following interesting corollary.

**Corollary 1** (Vanishing variance). *Under the same settings of those of Theorem 3, the variance of stochastic gradients sampled by SGD converges to zero as iteration $k$ goes to infinity.*

Thus, upon convergence, the common global minimizer structure in deep learning leads to a self-variance-reducing effect on SGD. Such a desirable effect is the core property of stochastic variance-reduced algorithms that reduces sample complexity Johnson & Zhang (2013). Hence, this justifies in part that SGD is a sample-efficient algorithm in learning deep models.

DISCUSSION

We want to mention that many nonconvex sensing models have an underlying true signal and hence naturally have common global minimizers, e.g., phase retrieval Zhang et al. (2017b), low-rank matrix recovery Tu et al. (2016), blind deconvolution Li et al. (2018), etc. This is also the case for some neural network sensing problems Zhong et al. (2017). Also, these problems have been shown to satisfy the so-called gradient dominance condition and the regularity condition locally around the global minimizers Zhou et al. (2016); Tu et al. (2016); Li et al. (2018); Zhong et al. (2017); Zhou & Liang (2017). These two geometric properties imply the star-convexity of the objective function, which necessarily imply the epochwise and iterationwise star-convex path of SGD. Therefore, our results also have implications on the convergence guarantee of SGD for solving these problems as well.

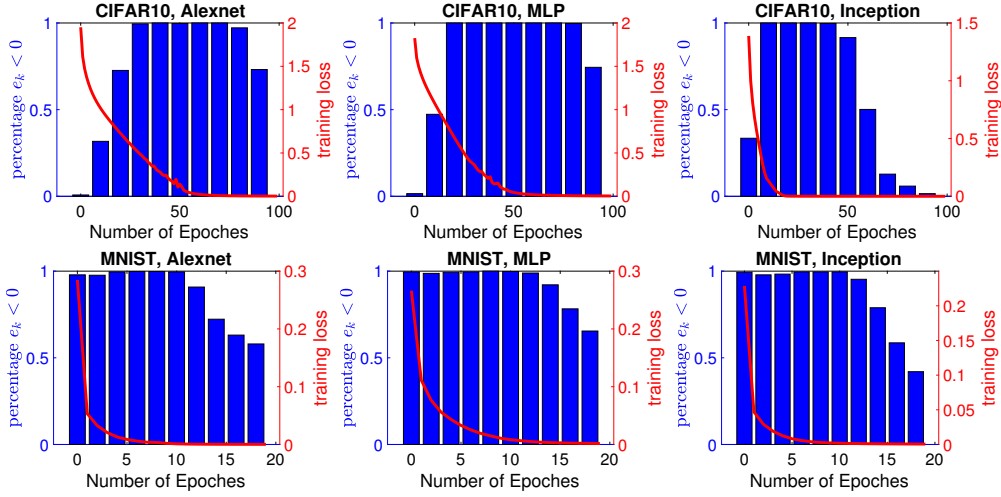

Figure 3: Verification of iterationwise star-convex path under crossentropy loss.

## 4.2 VERIFYING ITERATIONWISE STAR-CONVEX PATH OF SGD

In this subsection, we conduct experiments to validate the iterationwise star-convex path of SGD introduced in Definition 2.

We train the aforementioned three types of neural networks, i.e., MLP, Alexnet and Inception, on CIFAR10 and MNIST datasets using SGD. The hyperparameter settings are the same as those mentioned in Section 3.1. We train these networks for a sufficient number of epochs to achieve a near-zero training loss. Due to the demanding requirement for storage, we record the variable

sequence generated by SGD for all iterations in every tenth epoch, and set $x^*$ to be the final output of SGD. Then, for all the iterations in every tenth epoch, we evaluate the corresponding values of the terms on the left hand side in eq. (2) (denoted as $e_k$). Then, we report the fraction of number of iterations that satisfy the iterationwise star-convexity (i.e., $e_k < 0$) within such an epoch.

In all subfigures of Figure 3, the red curves denote the training loss and the blue bars denote the fraction of iterations that satisfy the iterationwise star-convexity within such an epoch. It can be seen from Figure 3 that, for all three networks on the MNIST dataset (second row), the path of SGD satisfies iterationwise star-convexity for most of the iterations, except in the last several epochs where the training loss (see the red curve) already well saturates at zero value. In fact, the convergence is typically observed well before such a point. This is because when the training loss is very close to the global minimum (i.e., the gradient is very close to zero), small perturbation of the landscape easily deviates the SGD path from the desired star-convexity. Hence, our experiments demonstrate that SGD follows the iterationwise star-convex path up to the convergence occurs. Furthermore, on the CIFAR10 dataset (first row of Figure 3), we observe a strong evidence for the iterationwise star-convex path of SGD after several epochs of the initial phase of training. This implies that the loss landscape on a more challenging dataset can be more complex.

Our empirical findings support the validity of the iterationwise star-convex path of SGD in a major part of practical training processes. Therefore, our convergence guarantee developed in Theorem 3 for SGD well justifies its practical success.

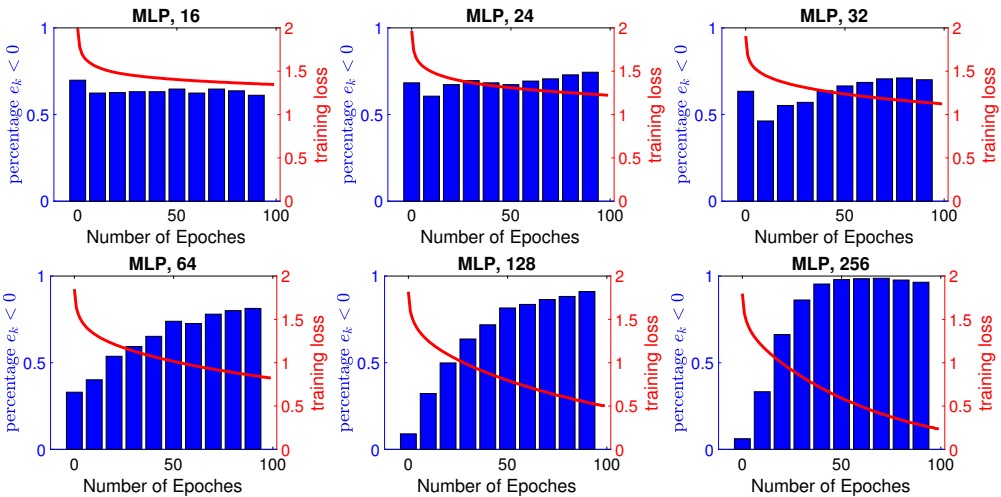

Figure 4: Iterationwise path on local minimum.

We next conduct further experiments to demonstrate that SGD follows the iterationwise star-convex path likely only for successful trainings to zero loss value, where a shared global minimum among all individual loss functions is achieved. To verify such a thought, we train an MLP using SGD on the CIFAR10 dataset under various settings with the number of hidden neurons ranging from 16 to 256. The results are shown in Figure 4, from which we observe that the training loss (i.e., red curves) converges to a non-zero value when the number of hidden neurons is small, implying that the algorithm likely attains a sub-optimal point which is not a common global minimum shared by all individual loss functions. In such trainings, we observe that the corresponding SGD paths have much fewer iterations satisfying the iterationwise star-convexity compared to the successful training instances shown in Figure 3. Thus, such empirical findings partially suggest that iterationwise star-convex SGD path more likely occurs when SGD can find a common global minimum, e.g., training overparameterized networks to zero loss value.

## 5 CONCLUSION

In this paper, we propose an epochwise star-convex property of the optimization path of SGD, which we validate in various experiments. Based on such a property, we show that SGD approaches a

global minimum at an epoch level. Then, we further examine the property at an iteration level, and empirically show that it is satisfied in a major part of training processes. As we prove theoretically, such a more refined property guarantees the convergence of SGD to a global minimum, and the algorithm enjoys a self-variance-reducing effect. We believe that our study sheds light on the success of SGD in training neural networks from both empirical aspect and theoretical aspect.

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

# Supplementary Materials

## A    PROOF OF THEOREM 1

Observe that the SGD update can be rewritten as the following optimization step

$$x_{k+1} = \underset{u \in \mathbb{R}^d}{\operatorname{argmin}} \Big\{ \underbrace{\ell_{\xi_k}(x_k) + \langle u - x_k, \nabla\ell_{\xi_k}(x_k)\rangle}_{f_{\xi_k}(u)} + \frac{1}{2\eta}\|u - x_k\|^2 \Big\}.$$

Note that the function $f_{\xi_k}(u)$ is linear, and we further obtain that for all $x^* \in \mathcal{X}^*_{\xi_k}$,

$$\begin{aligned}
\eta\big(f_{\xi_k}(x_{k+1}) - f_{\xi_k}(x^*)\big) &= \eta\langle\nabla\ell_{\xi_k}(x_k), x_{k+1} - x^*\rangle \\
&\overset{(i)}{=} \langle x_k - x_{k+1}, x_{k+1} - x^*\rangle \\
&= \frac{1}{2}\Big(\|x_k - x^*\|^2 - \|x_{k+1} - x^*\|^2 - \|x_{k+1} - x_k\|^2\Big),
\end{aligned}$$

where (i) uses the update rule of SGD. Rearranging the above inequality yields that

$$f_{\xi_k}(x_{k+1}) \le f_{\xi_k}(x^*) + \frac{1}{2\eta}\Big(\|x_k - x^*\|^2 - \|x_{k+1} - x^*\|^2 - \|x_{k+1} - x_k\|^2\Big). \tag{3}$$

On the other hand, by smoothness of the loss function, we obtain that

$$\begin{aligned}
\ell_{\xi_k}(x_{k+1}) &\le \ell_{\xi_k}(x_k) + \langle x_{k+1} - x_k, \nabla\ell_{\xi_k}(x_k)\rangle + \frac{L}{2}\|x_{k+1} - x_k\|^2 \\
&= f_{\xi_k}(x_{k+1}) + \frac{L}{2}\|x_{k+1} - x_k\|^2 \\
&\overset{(i)}{\le} f_{\xi_k}(x^*) + \frac{1}{2\eta}(\|x_k - x^*\|^2 - \|x_{k+1} - x^*\|^2) - (\frac{1}{2\eta} - \frac{L}{2})\|x_{k+1} - x_k\|^2 \\
&\overset{(ii)}{\le} f_{\xi_k}(x^*) + \frac{1}{2\eta}(\|x_k - x^*\|^2 - \|x_{k+1} - x^*\|^2), \tag{4}
\end{aligned}$$

where (i) follows from eq. (3) and (ii) is due to the choice of learning rate. Summing the above inequality over $k$ from $nB$ to $n(B+1) - 1$ yields that, for every $x^* \in \mathcal{X}^*$ in Definition 1,

$$\begin{aligned}
\|x_{n(B+1)} - x^*\|^2 &\le \|x_{nB} - x^*\|^2 - \sum_{k=nB}^{n(B+1)-1} 2\eta\big(\ell_{\xi_k}(x_{k+1}) - f_{\xi_k}(x^*)\big) \\
&\le \|x_{nB} - x^*\|^2 - \sum_{k=nB}^{n(B+1)-1} 2\eta\big(\ell_{\xi_k}(x_{k+1}) - \ell_{\xi_k}(x^*)\big), \tag{5}
\end{aligned}$$

where the last inequality follows from the star-convex path of SGD in Definition 1. The desired result follows from the above inequality and the fact that $\ell_{\xi_k}(x^*) = \inf_{u \in \mathbb{R}^d} \ell_{\xi_k}(u)$ for all $x^* \in \mathcal{X}^*$. Moreover, we conclude that the sequence $\{x_{nB}\}_B$ is bounded. By continuity of $\nabla\ell_i$ for all $i = 1, ..., n$ and the update rule of SGD, we further conclude that the entire sequence $\{x_k\}_k$ is bounded.

## B    PROOF OF THEOREM 2

We first collect some facts. Recall that $\mathcal{X}^* = \cap_{i=1}^n \mathcal{X}^*_i$ is non-empty and bounded. Consider any fixed $t \in \{0, \dots, n-1\}$ and recall that $k = nB + t$, $\xi_k = \pi_B(t+1)$. Then, one can check that the iterations $k$ with $\xi_k = v \in \{1, 2, ..., n\}$ form the subsequence $\{x_{nB+\pi_B^{-1}(v)-1}\}_B$.

Next, we prove item 1. Fix any $t \in \{0, \dots, n-1\}$ and sum eq. (4) over $k$ from $nB + t$ to $n(B+1) + t - 1$ yields that, for every $x^* \in \mathcal{X}^*$ in Definition 1,

$$\|x_{n(B+1)+t} - x^*\|^2 \le \|x_{nB+t} - x^*\|^2 - \sum_{k=nB+t}^{n(B+1)+t-1} 2\eta\big(\ell_{\xi_k}(x_{k+1}) - f_{\xi_k}(x^*)\big).$$

Further summing the above inequality over $B$ from $0$ to $K$ and rearranging, we obtain that

$$
\begin{aligned}
\|x_{n(K+1)+t} - x^*\|^2 \leq \|x_t - x^*\|^2 - \sum_{k=n}^{n(K+1)-1} 2\eta\big(\ell_{\xi_k}(x_{k+1}) - f_{\xi_k}(x^*)\big) \\
- \sum_{k=t}^{n-1} 2\eta\big(\ell_{\xi_k}(x_{k+1}) - f_{\xi_k}(x^*)\big) - \sum_{k=n(K+1)}^{n(K+1)+t-1} 2\eta\big(\ell_{\xi_k}(x_{k+1}) - f_{\xi_k}(x^*)\big), \\
\leq \|x_t - x^*\|^2 - \sum_{k=n}^{n(K+1)-1} 2\eta\big(\ell_{\xi_k}(x_{k+1}) - \ell_{\xi_k}(x^*)\big) \\
- \sum_{k=t}^{n-1} 2\eta\big(\ell_{\xi_k}(x_{k+1}) - f_{\xi_k}(x^*)\big) - \sum_{k=n(K+1)}^{n(K+1)+t-1} 2\eta\big(\ell_{\xi_k}(x_{k+1}) - f_{\xi_k}(x^*)\big),
\end{aligned}
\tag{6}
$$

where the last inequality follows from the the star-convex path of SGD in Definition 1. Consider the term $\ell_{\xi_k}(x_{k+1}) - \ell_{\xi_k}(x^*)$ in eq. (6) along the iterations with $\xi_k = v \in \{1, 2, ..., n\}$. Such term can be rewritten as $\ell_v(x_{nB+\pi_B^{-1}(v)}) - \ell_v(x^*)$. Suppose for certain $v \in \{1, 2, ..., n\}$ the sequence $\{\ell_v(x_{nB+\pi_B^{-1}(v)})\}_B$ does not converge to its global minimum $\inf_{x \in \mathbb{R}^d} \ell_v(x)$. Then, by the cyclic sampling scheme with reshuffle, we conclude that the first summation term in eq. (6) diverges to $+\infty$ as $K \to \infty$. Also, note that the last two summation terms have finite number of elements, which are all bounded as $\{x_k\}_k$ is bounded. Therefore, we conclude that the sequences $\{x_{nB+t}\}_B$ for $t = 0, ..., n-1$ converge to $x^*$ for all candidates $x^* \in \mathcal{X}^*$ in Definition 1. Next, consider the case in which there are multiple such candidate $x^*$s in Definition 1. Then, the previous sentence states that $\{x_{nB+t}\}_B$ converges to multiple limits, which cannot happen for a convergent sequence. This leads to a contradiction. Consider the other case that there is only one such candidate $x^*$ in Definition 1. Then, we conclude that all the sequences $\{x_{nB+t}\}_B$ for $t = 0, ..., n-1$ converge to $x^*$, i.e., the entire sequence $\{x_k\}_k$ converges to such unique common global minimizer. This contradicts with our assumption that $\{\ell_v(x_{nB+\pi_B^{-1}(v)})\}_B$ does not converge to $\inf_{x \in \mathbb{R}^d} \ell_v(x)$ for certain $v$. Combining both cases, we obtain the desired claim of item 1.

Next, we prove item 2. Note that sequence $\{x_k\}_k$ is bounded . Fix any $v \in \{1, ..., n\}$ and consider any limit point $z_v$ of $\{x_{nB+\pi_B^{-1}(v)}\}_B$, i.e., $x_{nB_j+\pi_{B_j}^{-1}(v)} \xrightarrow{j} z_v$ along a proper subsequence. From item 1 we know that $\ell_v(x_{nB_j+\pi_{B_j}^{-1}(v)}) \xrightarrow{j} \inf_{x \in \mathbb{R}^d} \ell_v(x)$. This, together with the continuity of the loss function, implies that $z_v \in \mathcal{X}_v^*$ for all $v$.

## C  PROOF OF THEOREM 3

Recall that eq. (4) shows that, for all $x^* \in \mathcal{X}_{\xi_k}^*$,

$$
\ell_{\xi_k}(x_{k+1}) \leq f_{\xi_k}(x^*) + \frac{1}{2\eta}\Big(\|x_k - x^*\|^2 - \|x_{k+1} - x^*\|^2\Big) \tag{7}
$$

$$
\overset{(i)}{\leq} \ell_{\xi_k}(x^*) + \frac{1}{2\eta}\Big(\|x_k - x^*\|^2 - \|x_{k+1} - x^*\|^2\Big), \tag{8}
$$

where (i) follows from the iterationwise star-convex path in Definition 2. Since $\ell_{\xi_k}(x_{k+1}) - \ell_{\xi_k}(x^*) \geq 0$, we conclude that for all $k = 0, 1, ...$ and every $x^* \in \mathcal{X}_{\xi_k}^*$,

$$
\|x_{k+1} - x^*\| \leq \|x_k - x^*\|. \tag{9}
$$

Next, consider any $v \in \{1, ..., n\}$, we show that every limit point $z_v$ of $\{x_{nB+\pi_B^{-1}(v)}\}_B$ is in $\mathcal{X}^*$. By eq. (9), we know that $\{\|x_{nB+\pi_B^{-1}(v)} - x^*\|\}_B$ is decreasing. Consider a limit point $z_v$ associated with the subsequence such that $x_{nB_j+\pi_{B_j}^{-1}(v)} \to z_v$. Then, for all $B_j \geq B$ we know that, for any fixed $x^* \in \mathcal{X}^*$,

$$
\|z_v - x^*\| \xleftarrow{j} \|x_{nB_j+\pi_{B_j}^{-1}(v)} - x^*\| \leq \|x_{nB+\pi_B^{-1}(v)} - x^*\|. \tag{10}
$$

Next, we prove by contradiction. Suppose that $z_v \notin \mathcal{X}^*$. Then, eq. (10) implies that $\|x_{nB+\pi_B^{-1}(v)} - x^*\| > 0$ for all large $B$ and any $x^* \in \mathcal{X}^*$. Combining this conclusion with item 2 of Theorem 2, it follows that all the limit points of $\{x_{nB+\pi_B^{-1}(v)}\}_B$ are in $\mathcal{X}_v^* \setminus \mathcal{X}^*$. Since $\mathcal{X}^* = \cap_{i=1}^n \mathcal{X}_i^*$, it follows that the limit points of $\{x_{nB+\pi_B^{-1}(u)}\}_B$ are different from those of $\{x_{nB+\pi_B^{-1}(v)}\}_B$ for any $u \neq v$. Now consider a subsequence $x_{nB_j+\pi_{B_j}^{-1}(v)} \to z_v \in \mathcal{X}_v^* \setminus \mathcal{X}^*$. Also, consider the subsequence $\{x_{nB_j+\pi_{B_j}^{-1}(u)}\}_j$ with $u \neq v$. By the cyclic sampling with reshuffle, we know that with probability one there is a subsequence $B_{j(s)}$ of $B_j$ such that $\pi_{B_{j(s)}}^{-1}(u) = \pi_{B_{j(s)}}^{-1}(v) + 1$ (this occurs with a constant probability in every epoch). Applying eq. (9) along this subsequence, we conclude that

$$\|x_{nB_{j(s)}+\pi_{B_{j(s)}}^{-1}(u)} - z_v\| \leq \|x_{nB_{j(s)}+\pi_{B_{j(s)}}^{-1}(v)} - z_v\| \xrightarrow{j} 0. \tag{11}$$

Let $j \to \infty$ in the above equation, we conclude that $x_{nB_{j(s)}+\pi_{B_{j(s)}}^{-1}(u)} \to z_v$, i.e., $z_v$ is a limit point of $\{x_{nB_j+\pi_{B_j}^{-1}(u)}\}_B$. Note that $z_v$ is a limit point of $\{x_{nB+\pi_B^{-1}(v)}\}_B$. This contradicts our previous conclusion that the limit points of $\{x_{nB+\pi_B^{-1}(u)}\}_B$ must be different from those of $\{x_{nB+\pi_B^{-1}(v)}\}_B$ for $u \neq v$. Thus, we must have for all $v = 1, \ldots, n$, every limit point $z_v$ of $\{x_{nB+\pi_B^{-1}(v)}\}_B$ is in $\mathcal{X}^*$. Then, by eq. (9) we further conclude that for all $B = 0, 1, \ldots$

$$\|x_{nB+\pi_B^{-1}(v)} - z_v\| \leq \|x_{n(B-1)+\pi_{B-1}^{-1}(v)} - z_v\|. \tag{12}$$

Note that $x_{nB_j+\pi_{B_j}^{-1}(v)} \xrightarrow{j} z_v$. Thus, for all $B \geq B_j$ the above inequality implies that

$$\|x_{nB+\pi_B^{-1}(v)} - z_v\| \leq \|x_{nB_j+\pi_{B_j}^{-1}(v)} - z_v\| \xrightarrow{j} 0. \tag{13}$$

This shows that $\{x_{nB+\pi_B^{-1}(v)}\}_B$ has a unique limit point $z_v$, which is an element of $\mathcal{X}^*$. Next, consider the limits $z_u, z_v (u \neq v)$ of the sequences $\{x_{nB+\pi_B^{-1}(u)}\}_B, \{x_{nB+\pi_B^{-1}(v)}\}_B$, respectively. By eq. (9) and the fact that $z_u \in \mathcal{X}^*$, we conclude that

$$\|x_{nB+\pi_B^{-1}(v)} - z_u\| \leq \|x_{n(B-1)+\pi_{B-1}^{-1}(v)} - z_u\| \leq \|x_{n(B-2)+\pi_{B-2}^{-1}(u)} - z_u\| \xrightarrow{B} 0.$$

Thus, $x_{nB+\pi_B^{-1}(v)} \to z_u$, and we conclude that $z_v = z_u$ for all $v \neq u$, i.e., the whole sequence $\{x_k\}$ has a unique limit point in $\mathcal{X}^*$.

## D  SUPPLEMENTARY EXPERIMENTS

In this section, we provide more experiments to illustrate the star-convexity property of the SGD path from other different aspects.

**Verification of epochwise star-convexity with different reference points**

In Figure 5 we verify the epochwise star-convexity of the SGD path by setting the reference point $x^*$ to be the output of SGD at different intermediate epochs (i.e., 60,80,100,120 epochs), where the SGD has already saturated close to zero loss. The experiments are conducted by training the Alexnet and MLP on Cifar10 using SGD. As can be seen from Figure 5, the epochwise star-convexity still hold (i.e., $e_B < 0$) after certain epochs in the initial training phase. This shows that the observed star-convex path does not depend on the choice of reference point (so long as they achieve near-zero loss).

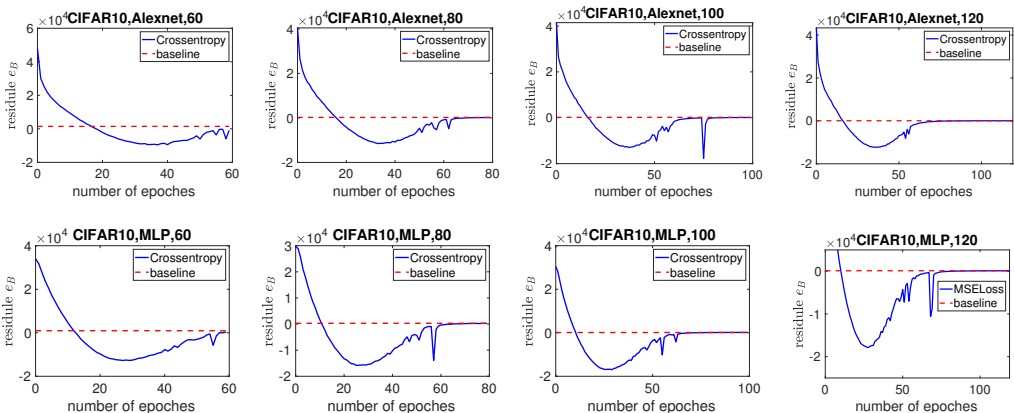

Figure 5: Verification of epochwise star-convexity when the reference point $x^*$ is taken at end of different epochs. Top plots correspond to the training of Alexnet with $x^*$ taken at 60th, 80th, 100th, and 120th epochs, and bottom plots correspond to the training of MLP with $x^*$ taken at 60th, 80th, 100th, and 120th epochs.

**Growth of weight norm in neural network training**

In Figure 6, we present the growth of the $\ell_2$ norm of network weights in training different neural networks on Cifar10 under the cross-entropy loss. It can be seen from the figure that the norm of the weights increases slowly (logarithmly) after the training loss achieves near-zero. This is because the gradient is nearly zero when the training is close to the global minimum, and therefore the updates of the weights are very small.

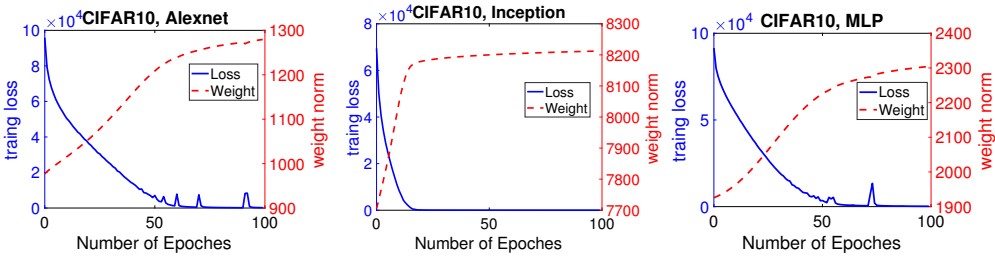

Figure 6: $\ell_2$ norm of network weights and training loss of different networks.

**Verification of star-convexity under MSE loss**

In Figure 7, we verify the epochwise star-convexity by training different networks on MNIST dataset under the MSE loss (i.e., $\ell_2$ loss). We note that unlike the cross-entropy loss, zero value can be achieved by the MSE loss. We set the reference point $x^*$ to be the output of SGD at the 40th epoch. It can be seen from the figure that the residue $e_B$ is negative along the entire optimization path, demonstrating that the SGD path satisfies the epochwise star-convexity under the MSE loss.

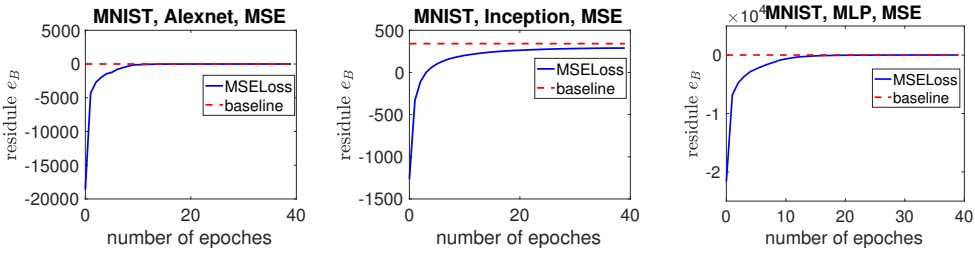

Figure 7: Verification of epochwise star-convexity under MSE loss (i.e., $\ell_2$ loss).

