# OpenReview forum: "SGD Converges to Global Minimum in Deep Learning via Star-convex Path"
_ICLR.cc/2019/Conference_

### Official Review · AnonReviewer3 · 2018-11-01
**Good theoretical paper**

**Rating:** 8
**Confidence:** 4

**Review:**

This paper analyzed the global convergence property of SGD in deep learning based on the star-convexity assumption. The claims seem correct and validated empirically with some observations in deep learning. The writing is good and easy to follow.

My understanding of the analysis is that all the claims seem to be valid when the solution is in a wide valley of the loss surface where the star-convexity holds, in general. This has been observed empirically in previous work, and the experiments on cifar10 in Fig. 2 support my hypothesis. My questions are:

1. How to guarantee the star-convexity will be valid in deep learning?
2. What network or data properties can lead to such assumption?

Also, this is a missing related work from the algorithmic perspective to explore the global optimization in deep learning:

Zhang et. al. CVPR'18. "BPGrad: Towards Global Optimality in Deep Learning via Branch and Pruning".

---

> ### Author Response · Authors · 2018-11-20
> **Response**
>
> We thank the reviewer for the valuable feedbacks.
>
> This paper aims at reporting an interesting star-convex property of the SGD optimization path that has been observed in training a variety of DL models, including MLP, CNN, residual networks and RNN (verified recently). Moreover, our theory is motivated by such a common observation and attempts to justify the role of this property plays in determining the convergence of the optimization in DL.
>
> Our response to the reviewer’s comments are provided as follows.
>
> 1. How to guarantee the star-convexity will be valid in deep learning?
>
> Response: We thank the reviewer for pointing out this question. It is definitely interesting to explore the underlying mechanism that leads to such a common observation.  We think that over-parameterization can be one of the important factors. We are currently investigating this issue theoretically on some simple networks, and our understanding so far favors such a direction.
>
> 2. What network or data properties can lead to such assumption?
>
> Response All our experiments are conducted on practical neural network training tasks with real datasets. From the experiments, we find that the property holds for a variety of network architectures (MLP, CNN, Inception, RNN) and different datasets (image, text, etc). We think that this can be an amenable property of over-parameterized network.
> In fact, several recent works ([1,2]) show that the optimization trajectories of SGD is generally smooth despite the nonconvexity and depth of the networks, and our star-convexity property can be viewed as another aspect that further promotes theoretical justification to deep learning optimization. We will explore these two questions more in future work.
>
> 3. There is a missing related work from the algorithmic perspective to explore the global optimization in deep learning:
> Zhang et. al. CVPR'18. "BPGrad: Towards Global Optimality in Deep Learning via Branch and Pruning".
>
> Response: We thank the reviewer for pointing out this interesting related work. We will cite this work in the upcoming revision.
>
> [1] Li et al. Visualizing the loss landscape of neural nets. To appear in NIPS 2018
> [2] Eliana Lorch. Visualizing deep network training trajectories with pca. In ICML Workshop on Visualization for Deep Learning, 2016.

---

### Official Review · AnonReviewer2 · 2018-11-01
**Interesting paper, but maybe less significant than it appears to be**

**Rating:** 5
**Confidence:** 5

**Review:**

This paper attempts to account for the success of SGD on training deep neural networks. Starting from two empirical observations: (1) deep neural networks can almost achieve zero training loss; (2) the path of iterates generated by SGD on these models follow approximately the “star convex path”, under the assumptions that individual functions share a global minima with respect to which the path of iterates generated by SGD satisfies the star convexity properties, the papers shows that the iterates converges to the global minima.

In terms of clarity, I think the paper can definitely benefit if the observations/assumptions/definitions/theorems are stated in a more formal and mathematically rigorous manner. For example:
- On page 3, “fact 1”: I don’t think “fact” is the right word here. “Fact” refers to what has been rigorously proved or verified, which is not the case for what is in the paper here. I believe “observation” is more appropriate. Also the assumption that l_i is non-negative should be formally added.
- On page 3, section 3.1: the x^* here is the last iteration produced by SGD. Then how can it be called the “global minima”? The caption of Figure 1 on page 4 is simply misleading.
- On page 4, the statement in definition 1 is more like a theorem than a definition. It is giving readers the impression that any path generated by SGD satisfies the star-convex condition, which is not the case here. A definition should look like “we call a path generated by SGD a star-convex path if it satisfies …”. Definition 2 on page 6 has the similar issue.

In terms of quality, while I believe the paper is technically correct,  I have one minor question here:
Page 3, Fact 1: How can you conclude that the set of common global minimizers are bounded? In fact I don’t believe this is true at all in general. If you have a ReLu network, you can scale the parameters as described in [1], then the model is invariant. Therefore, the set of common minimizer is definitely NOT bounded.

In terms of significance, I think this paper is very interesting as it attempts to draw the connection between the aforementioned observations and the convergence properties of SGD. Unfortunately I think that this paper is less significant than it has appeared to be, although the analysis appears to be correct.

First of all, all the analysis of this paper is based on one very important and very strong assumption, namely, all individual functions $l_i$ share at least one common global minimizer. The authors have attempted to justify this assumption by empirical evidences (figure 1). However, achieving near-zero loss is completely different from achieving exact zero because only when the model achieves exact zero can you argue that a common global minimizer exists.

Secondly, the claim that the iterate converges to the global minima is based on the assumption that the path follows an “epoch-wise star-convex” property. From this property, it only takes simple convex analysis to reach the conclusion of theorem 1 and 2. Meanwhile, the assumption that the path does follow the “epoch-wise start-convex” properties is not at all informative. It is not clear why or when the path would follow such a path. Therefore theorem 1 and 2 are not more informative than simply assuming the sequence converges to a global minimizer.

In fact, it is well-known that SGD with constant stepsize converges to the unique minimizer if one assumes the loss function F is strongly convex and the variance of the stochastic gradient g_k is bounded by a multiple of the norm-square of the true gradient:
Var(g_k) <= M ||∇F(x_k)||^2
Which is naturally satisfied if all individual functions share a common minimizer. Therefore, I don’t think the results shown in the paper is that surprising or novel.

With respect to the empirical evidence, the loss function l_i is assumed to be continuously differentiable with Lipschitz continuous gradients, which is not true for networks using ReLU-like activations. Then how can the paper use models like Alexnet to justify the theory? Also, if what the authors claim is true, then the stochastic gradient would have vanishing variance as it approaches x^*. Can the authors show this empirically?

In summary, I think this paper is definitely interesting, but the significance is not as much as it would appear.

Ref:
[1] Dinh, L., Pascanu, R., Bengio, S., & Bengio, Y. (2017). Sharp minima can generalize for deep nets. arXiv preprint arXiv:1703.04933.

---

> ### Author Response · Authors · 2018-11-20
> **Response**
>
> We thank the reviewer for the valuable feedbacks.
>
> This paper aims at reporting an interesting star-convex property of the SGD optimization path that has been observed in training a variety of DL models, including MLP, CNN, residual networks and RNN (verified recently). Moreover, our theory is motivated by such a common observation and attempts to justify the role of this property plays in determining the convergence of the optimization in DL.
>
> Our response to the reviewer’s comments are provided as follows.
>
> Quality:  In Fact 1: How can you conclude that the set of common global minimizers are bounded?
>
> A: We thank the reviewer for pointing out this. In fact, our theoretical results do not require that all the global minima be bounded. To be precise, we only need the star-convexity to hold for a bounded subset of the common global minimum, under which our theory guarantees that SGD converges to one of the elements in that set.  We clarify this in the revision.
>
> Clarity: On page 3, “fact” is the right word here. On Page 3 the x^* here is the last iteration produced by SGD. On page 4, the statement in definition 1 is more like a theorem.
>
> A: We thank the reviewer for valuable suggestions. In the revision, we use ``observation’’ instead of “fact”. We add the non-negativity assumption. We now refer to x^* as the output of SGD. We restate definition 1&2.
>
> Significance 1) The analysis of this paper is based on ...
>
> A: We have added new experiments to demonstrate that for the star-convexity of SGD holds for the MSE loss function (which can achieve zero loss).  We agree that cross-entropy achieves only near-zero loss, but such an approximation is not that unreasonable, as can be observed by the experiments that we added as Fig. 6 in Appendix D, which illustrates that the cross-entropy loss is nearly zero for certain finite norm of the weight parameters. Hence, we do expect that such approximation can convey useful information.
> We believe that we should not restrict ourselves only to theory that exactly matches what happens in practice. Near-zero loss is widely observed for training over-parameterized neural networks. Thus, the common minimizer assumption is motivated by this observation, and has led us to discover the star-convexity of SGD paths empirically, and further develop the convergence of SGD based on such a property. Hence, the approximate common global minimizer does yield consistent practical and theoretical results that explain what happens in deep learning.
>
> 2) Secondly, ….
>
> A: We first want to point out that the “epoch-wise star-convexity” is an accumulative effect of the residual error of every component loss over one epoch, which is a nontrivial and much weaker condition than that the entire loss function is star-convex over the points that SGD visits. Thus, assuming “epoch-wise star-convexity”, the proofs of theorems 1 and 2 do not follow the conventional convex analysis. One can of course argue that it is simple, but we think the focus here should be the information that it conveys in such a context.
> Second, we report a star-convex path property over a wide range of DL training tasks, which have not been reported in the existing literature to our knowledge. We do think this is an informative discovery. Of course, understanding when and why such a property hold for DL training is definitely important and deserves exploration in the future work.
>
> 3) In fact, it is well-known that SGD with constant step size …
>
> A: We want to point out the difference between our theory and the result mentioned by the reviewer. The star-convexity is much more relaxed than that the loss function F being strongly convex. Second, the bounded variance assumption that Var(g_k) <= M ||∇F(x_k)||^2 is hard to justify in general, and it is not clear to what extent can it be justified in DL tasks. Not to mention that it is clearly not true that the loss function is strong convex! In contrast, our star-convexity assumption is verified by various DL experiments as we report in the paper. Moreover, under strong convexity, traditional analysis only guarantees the convergence of the sequence in probability, which is much weaker than our deterministic convergence results in Theorem 3.
>
> 4) With respect to the empirical evidence, ...
>
> A: We thank the reviewer for pointing out this, and we are aware of it. We use ReLU activation as it is commonly used in DL tasks. Of course, one can use a smoothed version of ReLU (softplus) and obtain nearly the same result. We clarified this and added more experiments on this in the revision.
> For the variance, we do observe that the variance vanishes as SGD converges, and in fact report such a property as Corollary 1 in the originally submitted version. This is a necessary observation when SGD converges to a common global minimizer, and therefore also justifies the existence of common global minimum to some extent. We add experiments on this in the revision.

---

### Official Review · AnonReviewer1 · 2018-11-02
**The paper provides interesting idea but the empirical results may be biased due to ill-posed problem**

**Rating:** 6
**Confidence:** 4

**Review:**

The paper proposes a new approach to explain the effective behavior of SGD in training deep neural networks by introducing the notion of star-convexity. A function h is star-convex if its global minimum lies on or above any plane tangent to the function, namely h* >= h(x) + < h'(x), x*-x> for any x. Under such condition, the paper shows that the empirical loss goes to zero and the iterates generated by SGD converges to a global minimum. Extensive experiments has been conducted to empirically validate the assumption.

The paper is very well organized and is easy to follow. The star-convexity assumption is very interesting which provides new insights about the landscape of the loss function and the trajectory of SGD. It is in general difficult to theoretically check this condition so several empirical verifications has been proposed. My main concern is about these empirical verifications.

1) The minimum of the cross entropy loss lies at infinity
The experiments are performed respect to the cross entropy loss. However, cross entropy loss violates Fact 1 since for any finite weight, cross entropy loss is always strictly positive. Thus the zero is never attained and the global minimum always lies at infinity. As a result, the star-convexity inequality h* >= h(x) + < h'(x), x*-x> hardly makes sense since x* is at infinity and neither does the theorem followed.
In this case, a plot of the norm of xk is highly suggested since it is a sanity check to see whether the iterates goes to infinity.

2) The phenomenon may depend on the reference point, i.e last iterate
Since the minimum is never attained, the empirical check of the star-convexity maybe biased. More precisely, it might be possible that the behavior of the observed phenomenon depends on the reference point, i.e. the last iterate. Therefore, it will be interesting to see if the observed phenomenon still holds when varying the stopping time, for instance plot the star convexity check using the iterates at 60, 80, 100, 120 epochs as reference point.

In fact, the experiments shown in Figure 4 implicitly supports that the behavior may change dramatically respect to different reference point. The reason is that the loss in these experiments are far away from 0, meaning that we are far from the minimum, thus checking the star-convexity does not make sense because the star-convexity is only defined respect to the minimum.

Overall, the paper provides interesting idea but the empirical results may be biased due to ill-posed problem

---

> ### Author Response · Authors · 2018-11-20
> **Response**
>
> We thank the reviewer for the valuable feedbacks.
>
> This paper aims at reporting an interesting star-convex property of the SGD optimization path that has been observed in training a variety of DL models, including MLP, CNN, residual networks and RNN (verified recently). Moreover, our theory is motivated by such a common observation and attempts to justify the role of this property plays in determining the convergence of the optimization in DL.
>
> Our response to the reviewer’s comments are provided as follows.
>
> 1) The minimum of the cross entropy loss lies at infinity. It is a sanity to check whether the iterates goes to infinity.
>
>  Response: We thank the reviewer for pointing out this, and we are aware of it. We choose to present the results on cross entropy as it is widely used in DL applications. In fact, we verified the star-convexity property for training other losses that by nature can achieve zero such as the MSE loss (please see the experiment results that we added in Fig.7 in Appendix D in supplementary).
> Under the cross-entropy loss, we found that after the training loss is very close to zero, the l_2 norm of the corresponding iterate grows only logarithmically. This can be clearly seen from the experiments that we added in Fig.6 in Appendix D of supplementary. We further note that such a phenomenon has also been observed and justified in Fig.2 of [1]).  Thus, empirically, it is reasonable to treat the loss value to be approximately zero (i.e., reaches minimum) with a bounded weight norm.
> [1] ``The Implicit Bias of Gradient Descent on Separable Data’’, Soundry et al 2018
>
> 2) The phenomenon may depend on the reference point, i.e., last iterate
> Since the minimum is never attained, the empirical check of the star-convexity maybe biased. More precisely, it might be possible that the behavior of the observed phenomenon depends on the reference point, i.e. the last iterate. Therefore, it will be interesting to see if the observed phenomenon still holds when varying the stopping time, for instance plot the star convexity check using the iterates at 60, 80, 100, 120 epochs as reference point.
> In fact, the experiments shown in Figure 4 implicitly supports that the behavior may change dramatically respect to different reference point. The reason is that the loss in these experiments are far away from 0, meaning that we are far from the minimum, thus checking the star-convexity does not make sense because the star-convexity is only defined respect to the minimum.
>
> Response: As can be seen in the experiments that we added in Fig.5 in Appendix D of the supplementary materials, we have checked the star-convex property by taking the reference point at different intermediate iterates (60, 80, 100, 120) as the reviewer suggested. We found that star-convexity still holds under these choices of reference points, and therefore such a property does not depend on the choice of reference point so long as their loss are (nearly) zero. This observation is common for over-parameterized networks which can achieve near zero loss and therefore can have common global minimum.
> We emphasize that the reference point in our star-convexity must be the minimizer that achieves zero loss. Hence, in experiments, we must set the reference point at the epochs where the corresponding loss is nearly zero. The points at intermediate iterates with a high loss value cannot be chosen as reference point of star-convexity, because these points cannot be treated as the common global minimizer..
> Regarding the experiments in Figure 4, they are conducted on under-parameterized networks where (approximate) zero loss cannot be achieved, and the algorithm in fact does not find a common global minimum. This experiment is to justify the role of the over-parameterization (or the common global minimum) plays in determining the star-convex optimization path.

---

> > ### Comment · AnonReviewer1 · 2018-11-24
> > **Reply**
> >
> > I thank authors for carefully addressing my concerns and I raise my score accordingly.
> >
> > All the additional experiments are very interesting. The experiments of different reference point in Fig.5 suggests that the phenomenon of star-convexity is quite stable as long as the iterate are in a region "closed" to the minimum. The experiments on the norm of the iterate somehow supports this observation, since the change in the norm becomes less significant after some iteration, i.e. stableness of the last iterates.
> >
> > Moreover, according to Fig.5, it seems like a phase transition occurs after some iterations. In particular, the behavior of the first iterations are quite different from the latest ones, hence it would be interesting to develop some further characterization of such phase transition.

---

### Author Response · Authors · 2018-11-28
**Authors’ comment on very recent related studies**

While our submission is under review, a few related but very different studies [1,2,3] were posted on Arxiv very recently. We would like to briefly clarify our difference from these results here, which we will further include into the future version of this paper.

The studies [1,2,3] proved the optimization convergence results of gradient-based algorithms in over-parameterized deep learning based on various technical assumptions about overparameterization and algorithm parameters, which are still subject to validations in deep learning practice in the future. As a comparison, our proof of convergence is based on the star-convexity property, which we have verified in training a variety of practical neural networks on real datasets. The star-convexity property by itself is a new finding and can be of independent interest. Furthermore, our result is on the convergence of parameters, which in nature is different from the results in [1,2,3] on the convergence of the loss function value.

[1]``Gradient Descent Finds Global Minima of Deep Neural Networks’’, Simon S. Du, Jason D. Lee, Haochuan Li, Liwei Wang, Xiyu Zhai

[2] ``A Convergence Theory for Deep Learning via Over-Parameterization’’, Zeyuan Allen-Zhu, Yuanzhi Li, Zhao Song

[3] ``Stochastic Gradient Descent Optimizes Over-parameterized Deep ReLU Networks’’,
Difan Zou, Yuan Cao, Dongruo Zhou, Quanquan Gu.

---

### Meta-Review · Area_Chair1 · 2018-12-16
**New notion of nonconvexity**

**Confidence:** 3
**Recommendation:** Accept (Poster)

**Metareview:**

The proposed notion of star convexity is interesting and the empirical work done to provide evidence that it is indeed present in real-world neural network training is appreciated.  The reviewers raise a number of concerns. The authors were able to convince some of the reviewers with new experiments under MSE loss and experiments showing how robust the method was to the reference point. The most serious concerns relate to novelty and the assumptions that individual functions share a global minima with respect to which the path of iterates generated by SGD satisfies the star convexity property. I'm inclined to accept the authors rebuttal, although it would have been nicer had the reviewer re-engaged. Overall, the paper is on the borderline.